



# Diel Cycle Impacts on the Chemical and Light Absorption Properties of Organic Carbon Aerosol from Wildfires in the Western United States

Benjamin Sumlin[1], Edward Fortner[2], Andrew Lambe[2], Nishit Shetty[1], Conner Daube[2], Pai Liu[1], Francesca Majluf[2], Scott Herndon[2], and Rajan K. Chakrabarty[1]

[1]Center for Aerosol Science and Engineering, Department of Energy, Environmental, and Chemical Engineering, Washington University in St. Louis, St. Louis, Missouri
[2]Aerodyne Research, Inc., Billerica, Massachusetts

*Correspondence to*: Benjamin Sumlin (bsumlin@wustl.edu) and Rajan K. Chakrabarty (chakrabarty@wustl.edu)

**Abstract.** Organic aerosol (OA) emissions from biomass burning have been the subject of intense research in recent years, involving a combination of field campaigns and laboratory studies. These efforts have aimed at improving our limited understanding of the diverse processes and pathways involved in the atmospheric processing and evolution of OA properties, culminating in their accurate parameterizations in climate and chemical transport models. To bring closure between laboratory and field studies, wildfire plumes in the western United States were sampled and characterized for their chemical and optical properties during the ground-based segment of the 2019 Fire Influence on Regional to Global Environments and Air Quality (FIREX-AQ) field campaign. Using a custom-developed multiwavelength integrated photoacoustic-nephelometer (MIPN) spectrometer in conjunction with a suite of instruments, including an oxidation flow reactor equipped to generate hydroxyl (OH·) or nitrate (NO$_3$·) radicals to mimic daytime or nighttime oxidative aging processes, we investigated the effects of multiple equivalent days or nights of OH·/NO$_3$· exposure on the chemical composition and mass absorption cross-sections (MAC($\lambda$)) at 488 and 561 nm of OA emitted from wildfires in Arizona and Oregon. We found that OH· exposure reduced the wavelength-dependent MAC($\lambda$) by a factor of $0.72 \pm 0.08$, consistent with previous laboratory studies. On the other hand, NO$_3$· exposure increased it by a factor of up to $1.69 \pm 0.38$. The MAC enhancement following NO$_3$· exposure was correlated with an enhancement in CHO$_1$N and CHO$_{gt1}$N ion families measured with an aerosol mass spectrometer.

## 1 Introduction

Wildfires constitute one of the main contributions to the global atmospheric organic aerosol (OA) burden (Murphy et al., 2006; Zhang et al., 2007). The diverse physical, chemical, and optical properties of OA, the varied pathways involved in their formation in fires, and subsequent atmospheric processing introduce uncertainty into the analysis of their climate impacts, preventing extensive and accurate representation within regional and global scale models. Laboratory studies have, in the recent several years, demonstrated the effects of atmospheric processing on the optical, chemical, and physical properties of biomass burning OA (BBOA) or secondary organic aerosol (SOA) from biomass sources (Jimenez et al., 2009; Kroll et al., 2011; Lambe et al., 2013; Sumlin et al., 2017a). BBOA is typically also associated with brown carbon (BrC), a class of OA known to strongly absorb sunlight in the UV and short visible



wavelengths (Pósfai et al., 2004; Chakrabarty et al., 2010). Following diurnal cycles, the impacts of atmospheric oxidation can be broadly divided into daytime- and nighttime-driven processes. During daylight hours, the dominant oxidant that may influence OA properties is OH·. At night, when OH· production is significantly lower, $O_3$ or $NO_3·$ are the most likely oxidants to influence biomass burning OA (BBOA) properties. Initial laboratory studies suggest OH· and $NO_3·$-induced oxidative aging affect the light absorption properties of BBOA surrogates in markedly different

ways. OH· exposure to diminish light absorption by fragmenting large chromophoric molecules (Lambe et al., 2013; Sumlin et al., 2017a), while $NO_3·$ oxidation has been shown to enhance light absorption by adding nitrogen-containing chromophoric functional groups (Li et al., 2019; Cheng et al., 2020; Li et al., 2020; He et al., 2021).

     The need to study the consequences of such atmospheric processing on BBOA has motivated the development and use of environmental chambers capable of containing and controlling a sample of aerosol as it ages naturally (such as

a Teflon bag sitting in open sunlight) or by accelerating the aging process by exposing aerosols to increased concentrations of atmospheric oxidants, UV light, or both (Carter et al., 1995; Cocker et al., 2001; Grieshop et al., 2009; Cubison et al., 2011; Ortega et al., 2013). Such chambers are often large, on the order of several cubic meters (Cocker et al., 2001), and the relatively slow pace of a given experiment may suffer from unrealistic loss mechanisms including wall losses (including electrostatic, diffusional, and gravitational) (McMurry and Grosjean, 1985; Zhang et

al., 2014; Wang et al., 2018) and aerosol agglomeration and coagulation (Pierce et al., 2008). Recently, oxidation flow reactors (OFRs) have been used as a field-deployable alternative to traditional environmental chambers. OFRs are comparatively small, on the order of $0.01 \text{ m}^3$, with residence times on the order of minutes and oxidant concentrations that are 100-1000 times higher than typical ambient levels, resulting in equivalent atmospheric aging time scales from hours to weeks (Lambe et al., 2011).

While most work with OFRs has focused on OH·-initiated oxidative aging processes , new methods are available to employ $NO_3·$ in field-based OFR studies (Lambe et al., 2020; Li et al., 2020) that expand on previous work (Palm et al., 2017). The consequences of such atmospheric processing over multiple diurnal cycles and broad geographical areas introduce significant errors into the aerosol components of radiative transfer models and make it difficult to constrain the impacts that wildfires and other large-scale aerosol events will have on the climate, especially in areas

where air quality is expected to suffer under the effects of anthropogenic climate change (McClure and Jaffe, 2018).Following previous laboratory studies that investigated the various properties of BrC generated from smoldering biomass (Sumlin et al., 2017b; Sumlin et al., 2018a; Sumlin et al., 2018b), including their optical characteristics following OH· exposure in an OFR , this work investigates the effects of daytime and nighttime aging on ambient BrC properties during the Fire Influence on Regional to Global Environments and Air Quality (FIREX-AQ) campaign, an

interagency mission led by NASA and NOAA, conducted during the wildfire season of 2019 (Warneke, 2018). This work focuses on four OFR-based BrC oxidative aging experiments performed in August 2019 at ground level near the North Rim of the Grand Canyon in Arizona during the Castle and Ikes fires, and in Eastern Oregon, during the 204 Cow Fire. Results are used to quantify the effects of diurnal cycle-driven oxidation processes on the BrC refractive index.





## 2 Methods

### 2.1 Wildfire emissions sampled during OFR experiments

### 2.1.1 Castle and Ikes Fires, Northern Arizona

At the time of the FIREX-AQ campaign, two large fires were burning in close proximity to each other near the North Rim of the Grand Canyon, near Page, AZ. These fires, the Castle and Ikes fires (36.51° N, 112.28° W, and 36.35° N, 112.29° W, respectively), were ignited by lightning on 12 and 25 July 2019, respectively, by lightning strikes. Since these fires threatened no structures or other property and were burning through a forest densely packed with litter, underbrush, and dead fallen trees, the fires were allowed to burn within prescribed limits to serve their role as the forest's natural mechanism of cleanout and renewal while fire crews in the area used Castle and Ikes as training opportunities.

The Castle and Ikes fires burned near the Oquer Canyon in the Kaibab National Forest, a densely forested area consisting predominantly of Ponderosa pine (*Pinus ponderosa*), Douglas fir (*Pseudotsuga menziesii*), Englemann spruce (*Picea engelmannii*), and quaking aspen (*Populus tremuloides*). Minor species include Gambel oak (*Quercus gambelii*), and various shrub and grass species of sagebrush (*Artemesia*) and bitterbrush (*Purshia*). By 20 August 2019, the fire management areas had grown to approximately 24,000 acres (~97 km$^2$) combined. The Aerodyne Mobile Laboratory (AML, described below) approached from Page via Highway 89A and State Route 67. An OH·-OFR experiment (OH$_{Arizona}$) was conducted on 20 August at approximately 17:45 UTC (10:45 MST) at 36.61°N, 112.19°W (Fig. 1), and a NO$_3$·-OFR experiment (NO$_{3,Arizona}$) was conducted on 22 August at approximately 10:30 UTC (03:30 MST) at 36.54° N, 112.17° W (Fig. 2).

[Figure 1]

[Figure 2]

### 2.1.2 204 Cow Fire, Eastern Oregon

After returning from Arizona, the nearest fire of interest was the 204 Cow Fire (44.28° N, 118.47° W) in the Malheur National Forest region of the Blue Mountains in eastern Oregon, approximately 200 km west-southwest of McCall. The 204 Cow Fire was ignited on 9 August by lightning and grew to burn 9,668 acres (approximately 29 km$^2$) until its containment on 15 October. The region is predominantly forested by various species of pine, fir, and juniper trees (*Pinus*, *Abies*, and *Juniperus*, respectively) along with shrub species of sagebrush (*Artemesia*). The fire had burned approximately 5,500 acres (22.3 km$^2$) by 26 August, when both OH·-OFR ("OH$_{Oregon}$", 09:30 UTC, 03:30 MDT, 44.23° N, 118.40° W) and NO$_3$·-OFR ("NO$_{3,Oregon}$", 12:30 UTC, 06:30 MDT, 44.25° N, 118.40° W) experiments took place (Fig. 3). The AML approached from State Route 26 and then traversed a series of National Forest Development roads to conduct sampling less than 3 km from the fire management area.

[Figure 3]





The AML, a mobile sampling platform equipped with a suite of research-grade instrumentation, was used to facilitate
the in situ OFR-based experiments described here. The AML travelled throughout Idaho, Oregon, Utah, and Arizona
during FIREX-AQ, sampling continuously when not conducting OFR experiments. The general sampling strategy
was to search for smoke-filled valleys and transect plumes with the AML, using Tuneable Infrared Laser Direct
Absorption Spectrometer (TILDAS, Aerodyne Research, Inc.) (McManus et al., 2011a; McManus et al., 2011b)
measurements of hydrogen cyanide (HCN) as a tracer for biomass smoke plumes (Li et al., 2000). Upon identification
of a suitable location, the AML parked with the sample inlet on the front of the truck facing into the wind to avoid
self-sampling of its own exhaust. A $PM_{2.5}$ cyclone with a small mesh screen to filter out extremely large ash particles
was attached to the inlet. A schematic of the OFR experimental setup is given in Fig. 4, and Section 2.2 discusses
instrumentation specifically relevant to the OFR-based field measurements in more detail.

AML positioning, including latitude, longitude, and altitude was measured by a Vector V103 GPS Compass
(Hemisphere GNSS, Inc., Scottsdale, AZ). Wind velocity was measured by an ultrasonic anemometer (model 86000,
R. M. Young Company, Traverse City, MI). Additional meteorology products were obtained from the NOAA Air
Resources Laboratory North American Mesoscale 12 km Archive, and plume ages were approximated using the
Hybrid Single-Particle Lagrangian Integrated Trajectory (HYSPLIT) model (Stein et al., 2016). During FIREX-AQ,
aerosol optical properties were measured using a novel Multiwavelength Integrated Photoacoustic-Nephelometer
(MIPN) spectrometer described in more detail in Section 2.2.1. Aerosol composition and mixing state properties were
measured with a Soot Particle Aerosol Mass Spectrometer (SP-AMS, Aerodyne Research, Inc., Onasch et al., 2012)
and Single Particle Soot Photometer (SP2 Droplet Measurement Technologies, (Stephens et al., 2003) ). A Vocus
Proton Transfer Reaction Mass Spectrometer (Krechmer et al., 2018) (PTR-MS, Aerodyne Research, Inc./Tofwerk
AG, Switzerland ) sampled VOCs.

115                                                             **[Figure 4]**

## 2.2 Instrumentation

### 2.2.1 Multiwavelength Integrated Photoacoustic-Nephelometer (MIPN)

A description of the design and development of this instrument, including calibration procedures, has been given
previously (Arnott et al., 1999; Arnott et al., 2000; Moosmüller and Arnott, 2003; Varma et al., 2003; Lewis et al.,
2008). Briefly, the aerosol light absorption coefficient ($\beta_{abs}$, $Mm^{-1}$) was measured with the prototype version of a new
Multiwavelength Integrated Photoacoustic-Nephelometer (MIPN) at two wavelengths ($\lambda$ = 488 and 532 nm). The
MIPN is based upon single-wavelength instruments constructed at Washington University in St. Louis and described
previously (Sumlin et al., 2017a; Sumlin et al., 2018a; Sumlin et al., 2018b). Novel to the MIPN is a dual-cell
arrangement wherein the sample stream is split and one branch is filtered to remove particulate matter, thereby
simultaneously sampling a particle-free gaseous background to account for noise from various sources such as ambient
acoustic, electrical, and flow noise. Data were acquired from each wavelength in serial at 0.5 Hz for one minute per
wavelength and averaged across each wavelength's one-minute cycle. Due to a field malfunction, scattering
measurements were not obtained.



OA mass absorption cross-sections (MAC($\lambda$), $m^2$ $g^{-1}$) at the two operating wavelengths were calculated from the ratio of $\beta_{abs}$ measured with MPIN and OA mass concentrations measured with the SP-AMS after correcting for dilution and wall losses in the OFR. We additionally calculated the MAC($\lambda$) enhancement, $E_{MAC}(\lambda)$, from the ratio of OFR-processed and ambient MAC($\lambda$) values. Thus, $E_{MAC}(\lambda) > 1$ indicates an oxidative aging-induced absorption enhancement, while $E_{MAC}(\lambda) < 1$ indicates diminished absorption enhancement. To obtain reasonable estimates for

$E_{MAC}(\lambda)$ values that are comprehensive of a given oxidation step yet still capture the nature of the dynamic plume, $E_{MAC}(\lambda)$ was calculated using the average of aerosol MAC($\lambda$) during a given oxidation step divided by the average of the ambient steps immediately prior and after.

### 2.2.2 Aerosol Mass Spectrometer

The SP-AMS is a standard Aerodyne high resolution time of flight aerosol mass spectrometer (HR-ToF-AMS) with

an intracavity, CW laser vaporizer (Onasch et al., 2012). The AMS was operated to provide online chemically-speciated mass and sizing measurements of both non-refractory and refractory particles between approximately 70 – 2500 nm in aerodynamic diameter. A $PM_{2.5}$ inlet lens was installed on the AMS for this study, extending the range of 100% transmission efficiency of particles through the lens up to 2.5 µm in diameter. The SP-AMS laser was operated with approximately a 50% duty cycle. When the laser was off, the system was operated as a conventional AMS.

During $OH_{Arizona}$, $OH_{Oregon}$, and $NO_{3,Oregon}$, the SP-AMS was used. During $NO_{3,Arizona}$, the conventional AMS was used. The instrument was run with 20 second time resolution, and data points included both chemical speciation and mass loading by mass spectral analysis and particle sizing by species for each data point. In the SP-AMS, particles containing refractory materials (i.e. BC and many metals) are vaporized with a 1064 nm laser. The resulting vapor is ionized via electron impact and detected with the HR-TOF-AMS. In addition to the SP-AMS vaporization, the

conventional AMS heater (a heated tungsten surface at 600 °C, (Jayne et al., 2000; Canagaratna et al., 2007) was also used to measure the composition of any non-refractory particles. When the instrument was run as a conventional AMS, this was the only heater used.

### 2.2.3 Potential Aerosol Mass (PAM) OFR

The PAM OFR (Aerodyne Research, Inc.; Lambe et al., 2011) is a horizontal 13 L aluminum cylindrical chamber (46

cm long × 22 cm ID) operated in continuous flow mode. Irradiance, relative humidity and temperature in the OFR were measured at the exit flange with UV (TOCON GaP6, sglux) and RH/T (SHT21, Sensiron) sensors, and ozone concentrations were measured with an ozone analyzer (Model 106-M, 2B Technologies).

### 2.3 Experiment Design

The total instrument plus makeup flowrate through the OFR was 6.4 liter $min^{-1}$, corresponding to a calculated mean

plug flow residence time of 123 sec. OFR experiments lasted 1 to 2 hours, during which the reagent inputs were controlled in steps to simulate varying degrees of atmospheric aging. At each step, instruments first sampled ambient air for 5-10 min (depending on the experiment timeline) while LFR/OFR conditions equilibrated, during which 5 L $min^{-1}$ of makeup flow was pulled through the OFR to reduce stagnation time. Then, an electronically actuated 3-way



valve was switched to connect the instruments to the OFR, and OFR-processed air was sampled for 5 to 10 min. After
each step, the OH· or NO₃· exposure was changed, and the above measurements were repeated. After each experiment,
the OFR was cleaned out by setting both sets of lamps to maximum output and overblowing the inlet with humidified
zero air until AMS measurements of background organic mass were below 0.3 μg m⁻³. To account for dilution and
particle wall losses in the OFR refractory black carbon (rBC) monitored with the SP2 was used as a chemically
conserved tracer. During all experiments, rBC accounted for ~ 2% to 5% of total aerosol mass (Figs. S1 and S2).
In the following sections, ambient steps are denoted "ambient_X" and oxidation steps are denoted "OFR_OH_X" for
OH· experiments, and for NO₃· experiments, the ozone-only step is denoted "NO3PAM_O3" and NO₃· oxidation steps
are denoted "NO3PAM_NO3_X". "X" indicates the step number. Each experiment includes a "background" step,
where ambient air was sampled through the dark OFR without oxidant generation.

### 2.3.1 NO₃ Experiment Design and Analysis

To generate NO₃·, N₂O₅ was first generated in the gas phase from the reaction NO₂ + O₃ → NO₃· + O₂ followed by
the reaction NO₃· + NO₂ → N₂O₅ in a 152.4 cm long x 2.22 cm ID perfluoroalkoxy laminar flow reactor (LFR) coupled
to the OFR (Lambe et al., 2020). Separate flows containing NO₂ (1% in N₂, Praxair) and O₃ were added to the LFR.
In these experiments, the NO₂ +N₂ flow rate was set between 0 and 40 cm³ min⁻¹, and O₃ was generated by passing
1.8 L min⁻¹ of O₂ through an ozone chamber housing a mercury fluorescent lamp (GPH212T5VH, Light Sources,
Inc.). The O3 mixing ratio that was input to the LFR was approximately 250 ppmv during NO₃·-OFR experiments.
The NO₂ + N₂ and O₂ flow rates were set using mass flow controllers. The N₂O₅ generated in the LFR thermally
decomposed at room temperature inside the OFR to generate NO₃·. The first oxidation step of NO₃·-OFR experiments
was with ozone only ("NO3PAM_O3") to assess the effect of O₃ exposure on BrC composition and optical properties
relative to ambient BrC. During NO₃,Arizona, NO₂ was stepped down from 40 to 20, 5, and 3 cm³ min⁻¹ to generate the
various oxidation time scales (Fig. S3). During NO₃,Oregon, inputs of 15 and 5 cm³ min⁻¹ were used. The integrated NO₃
exposure, defined as the product of the average NO₃· concentration and the mean OFR residence time ($\tau_{OFR}$), was
calculated using an estimation equation developed by Lambe et al. (2020):

$$\log[(NO_3)_{exp}] = a + b\log[273.15 + T_{OFR}] + c\log[\tau_{OFR}]$$
$$+ d\log[NO_2]_{0,LFR} + e\log[O_3]_{0,LFR} \times T_{OFR} + f\log[k_{w_{OFR,N_2O_5}}]$$
$$+ \log\left(\frac{[NO_2]_{0,LFR}}{[O_3]_{0,LFR}}\right) \times \left(g\left(\log[O_3]_{0,LFR}\right)^2 + h\log[O_3]_{0,LFR}\right)$$
$$- \frac{[NO_2]_{0,LFR}}{[O_3]_{0,LFR}} \times \left(i + j\log[O_3]_{0,LFR}\right) + k\log(NO_3R)_{ext}$$
$$+ l\log[NO_2]_{0,LFR} \times T + m\log[O_3]_{0,LFR} \times \log k_{w_{OFR,N_2O_5}} \tag{1}$$

Where a through m are empirical fit coefficients derived by Lambe et al. (2020) and summarized in Table S1, $T_{OFR}$ is
the measured temperature in the OFR, $[NO_2]_{0,LFR}$ and $[O_3]_{0,LFR}$ were the NO₂ and O₃ mixing ratios input to the LFR
(ppmv), $k_{wOFR,N2O5} = 0.01$ s⁻¹ is the assumed N₂O₅ wall loss rate coefficient in the OFR, and $(NO_3R)_{ext}$ is the external
NO₃· reactivity (s⁻¹), which was calculated from the summed products of ambient VOC concentrations (measured with
Vocus) and their and their NO₃· rate coefficients (found in the data repository, see Data Availability section).





Corresponding calculated NO$_3\cdot$ exposures ranged from $1.4\times10^{14}$ to $3.4\times10^{14}$ molec cm$^{-3}$ s, or approximately 3 to 8 equivalent days of atmospheric oxidation at a 24-hour average NO$_3\cdot$ concentration of $5\times10^8$ molec cm$^{-3}$ (Atkinson, 1991).

### 2.3.2 OH· Experiment Design

OH· was generated via photolysis of ambient O$_2$ and H$_2$O at $\lambda = 185$ nm plus photolysis of O$_3$ (generated from at $\lambda = 254$ nm using low-pressure mercury (Hg) lamps):

$H_2O + h\nu_{185} \rightarrow H + OH\cdot$

$H + O_2 \rightarrow HO_2$

$O_2 + h\nu_{185} \rightarrow 2O(^3P)$

$O(^3P) + O_2 \rightarrow O_3$

$O_3 + h\nu_{254} \rightarrow O_2 + O(^1D)$

$O(^1D) + H_2O \rightarrow 2OH\cdot$

A fluorescent dimming ballast was used to regulate current applied to the lamps (GPH436T5VH/4, Light Sources, Inc.). The dimming voltage applied to the ballast ranged from 1.6 V to 10 V direct current (DC). To extend the range of OH· concentrations below what is achievable with one set of lamps at 1.6 VDC, a second set of GPH436T5VH/4 lamps with added segments of opaque heat shrink tubing applied to ~86 % of the arc length (Rowe et al., 2020) was used. The OH$_{Arizona}$ OFR experiment began with the non-attenuated lamps set to maximum output, then switching to the attenuated lamps and stepping the dimming voltage from 10.0 to 5.0, 3.0, 2.0, and 1.6 V (Fig. S4). The OH$_{Oregon}$ experiment started with the non-attenuated lamps at 10V, then stepping the dimming voltage from 10.0 to 5.0 and 1.6 V. The OH exposure was calculated using Eq. (2) from Rowe et al. (2020):

$$\log[OH_{exp}] = \left(a + \left(b - c \times OHR^d_{ext} + e \times \log[O_3] \times OHR^f_{ext}\right) \times \log[O_3] + \log[H_2O]\right) + \log\left(\frac{\tau}{124}\right) \quad (2)$$

Where $a$ through $f$ are fit coefficients tabulated in Table S2, O$_3$ is the ozone mixing ratio measured at the exit of the OFR (molec cm$^{-3}$), OHR$_{ext}$ is the external OH· reactivity (s$^{-1}$), which was calculated from the summed products of ambient VOC concentrations (measured with Vocus) and their and their OH· rate coefficients (see Data Availability section), H$_2$O is the ambient water vapor mixing ratio (%), and $\tau$ is the residence time in the OFR. Corresponding calculated OH· exposures ranged from $3.91\times10^{11}$ to $2.53\times10^{12}$ molec cm$^{-3}$ s, or approximately 3 to 20 equivalent days of atmospheric oxidation at a 24-hour average OH· concentration of $1.5\times10^6$ molec cm$^{-3}$ (Mao et al., 2009).

## 3 Results and Discussion

### 3.1 Ambient Aerosol MAC(λ)

Table 1 summarizes the average MAC(λ) values of ambient aerosol obtained from Arizona and Oregon. The absorption Ångström exponent (AAE), which parameterizes the wavelength dependence of absorption and is calculated from the two-parameter formula






$$AAE(488\,\mathrm{nm}, 561\,\mathrm{nm}) = -\frac{\ln\left[\frac{\beta_{abs}(488\,\mathrm{nm})}{\beta_{abs}(561\,\mathrm{nm})}\right]}{\ln\left[\frac{488\,\mathrm{nm}}{561\,\mathrm{nm}}\right]}$$

(3)

was $0.83 \pm 0.25$ in Arizona and $0.91 \pm 0.21$ in Oregon.

**[Table 1]**

### 3.2 NO₃· Oxidative aging of BrC

Equivalent nighttime oxidation of between $3.28 \pm 0.00$ to $7.67 \pm 0.17$ days was performed across the $NO_{3,Arizona}$ and $NO_{3,Oregon}$ OFR experiments. $E_{MAC}(\lambda)$ is shown in Fig. 5 and summarized in Table 2, as the Oregon data points are obscured by their similarity. $E_{MAC}(\lambda)$ from $NO_{3,Arizona}$ increases by a factor of up to $1.69 \pm 0.38$ as a function of $NO_3$· exposure, as does $E_{MAC}(\lambda)$ from $NO_{3,Oregon}$, although to a lesser extent because $E_{MAC}(\lambda)$ was higher at lower equivalent time scales. There was a marked difference in $NO_{3,EXT}$ in Arizona and Oregon. In Arizona, calculated $NO_{3,EXT}$ values

ranged from 11.6 to 17.6 $s^{-1}$, while in Oregon, $(NO_3R)_{ext}$ 161.0 to 187.1 $s^{-1}$; the $NO_3$· reactivity of catechol alone was 48 to 56 $s^{-1}$. This indicates the presence of high concentrations of biomass burning VOCs (BBVOCs)that were highly reactive to $NO_3$·.

The differences in external reactivities manifested in different $E_{MAC}(\lambda)$ behavior at the two sites. On the shortest oxidation time scale in Arizona ($3.28 \pm 0.00$ equivalent nights), $E_{MAC}(488\,\mathrm{nm})$ was $0.99 \pm 0.26$ and $E_{MAC}(561\,\mathrm{nm})$

was $1.15 \pm 0.79$. This is in contrast to Oregon, where at $3.53 \pm 0.28$ equivalent nights, $E_{MAC}(488\,\mathrm{nm})$ was $1.47 \pm 0.01$ and $E_{MAC}(561\,\mathrm{nm})$ was $1.46 \pm 0.01$, which we attribute to the variations in BBVOC concentrations between sites. At longer time scales in Arizona, $E_{MAC}$ continued to increase, up to $E_{MAC}(488\,\mathrm{nm})$ of $1.69 \pm 0.38$ and $E_{MAC}(561\,\mathrm{nm})$of $1.29 \pm 0.84$. These values were commensurate with results from Oregon at only $3.87 \pm 0.40$ equivalent nights. Regardless of choice of site, $E_{MAC}(\lambda)$ consistently increased with increasing $NO_3$· oxidation time scales. Li et al. (2020)

previously observed this absorption enhancement effect in the laboratory by subjecting biomass burning BrC proxy aerosol derived from wood tar to a similar experimental setup to what was used here. They observed $E_{MAC}(\lambda)$ in the near-UV (330-400 nm) of approximately 2.4, and approximately 6.0 in the visible (400-550 nm), though this high value comes from the ratio of two relatively low $MAC(\lambda)$ values of 0.6 and 0.1 $m^2\,g^{-1}$, much lower than the $MAC(561\,\mathrm{nm})$ reported here, which, for ambient aerosol was $1.71 \pm 0.32$ in Arizona and $1.72 \pm 0.05$ in Oregon.

**[Figure 5]**

**[Table 2]**

A breakdown of the chemical speciation from $NO_{3,Arizona}$ was obtained from the AMS data from before and after the NO3PAM_NO3_1 step (refer to Fig. S3), the first oxidation step after ozonolysis. During this step, 217 ppm $NO_2$ was flowing into the LFR. $NO_{3,EXT}$ before and after this step was 17.56 and 13.90 $s^{-1}$, respectively, for an estimated

equivalent age of $5.76 \pm 0.22$ nights.. The individual ions measured by the AMS can be visualized for oxidative enhancement or depletion on a log-log scatterplot, shown with markers in Fig. S5, and with individual $m/z$ in Fig. S6. Ions above the solid black line are enhanced following $NO_3$· exposure, whereas ions below are depleted. The 2:1 and 1:2 lines are also included. Colors are per-ion family.


While knowledge of the individual *m/z* enhancement and depletion may inform future investigations, it is perhaps more illustrative to consider ion families writ large. Figure 6 shows the enhancement and depletion of the ion families in Figs. S5 and S6 on the basis of total ion mass in that family. Enhancement is calculated by summing the relative abundances of all ions within a family and taking the ratio of oxidized to ambient, similar to how $E_{MAC}(\lambda)$ is calculated.

**[Figure 6]**

The $CHO_{gt1}N$ ("$O_{gt1}$" indicates more than one oxygen atom in the molecule) family is enhanced by 211% and $CHO_1N$
("$O_1$" indicates a single oxygen atom in the molecule) by 132% through the PAM, with a corresponding diminishment in $CHO_{gt1}$ and $CHO_1$ to 79% and 91%, respectively. This does not imply that all $CHO_1N$ species come from $CHO_1$ (or $CHO_{gt1}N$ from $CHO_{gt1}$, similarly). The addition of nitrogen-containing functional groups is likely the cause of the observed light absorption enhancement, since nitrogenated aromatic hydrocarbons form during reactions with $NO_3 \cdot$ (Li et al., 2019) and can act as chromophores, increasing light absorption (Jacobson, 1999; Laskin et al., 2015; Xie et
al., 2019).

### 3.2.1 $O_3$ Oxidation Effects

The first step in the $NO_3 \cdot$ experiments was an $O_3$-only oxidation experiment in case the effects of ozonolysis on biomass burning smoke were significant. Since the $NO_3 \cdot$ oxidation steps carry a significant amount of residual $O_3$, the contribution of $O_3$ to changes in absorption behavior should be quantified and treated separately from $NO_3 \cdot$. Li et al.
(2020) performed a similar experiment in their laboratory oxidation studies and found that $NO_3 \cdot$ oxidation was the dominant chemical transformation mechanism when compared to $O_3$. They further reported a slight decrease in MAC(405 nm) quantified by a reduction in the imaginary refractive index from $0.017 \pm 0.005$ to $0.011 \pm 0.003$. However, differences in how Li et al. performed their experiments (by exposing aerosol to 35 ppbv $O_3$ for 10.6 hours) preclude any direct comparison to the short $O_3$ exposure time scales performed during FIREX-AQ.
$E_{MAC}(\lambda)$ due to $O_3$ during $NO_{3,Arizona}$ was $1.06 \pm 0.96$ and $1.05 \pm 0.94$ for 488 and 561 nm, respectively, indicating that $O_3$ effects during the PAM experiments were negligible, or at least dominated by $NO_3 \cdot$.

### 3.3 Daytime Oxidation

Compared to the $NO_3 \cdot$ experiments, a much broader range of equivalent atmospheric aging was mimicked in the $OH \cdot$ experiments. $NO_3 \cdot$ is a comparatively selective oxidant, while $OH \cdot$ tends to oxidize nearly all organic aerosol it comes
in to contact with. Between approximately 3 and 20 equivalent days of oxidation were performed. As expected, $E_{MAC}(\lambda)$ for these experiments nominally showed light absorption diminishment for most experiments, however, $E_{MAC}(\lambda)$ exceeded unity between $3.79 \pm 0.05$ and $17.41 \pm 4.04$ days. This effect of initial darkening and subsequent lightening has been reported previously (Sumlin et al., 2017a; Hems et al., 2020). The nature of this absorption enhancement mechanism was outside the scope of Sumlin et al. (2017) who rather focused on the likely mechanisms
of BrC light diminishment, however Hems et al. (2020) posit that the increase in absorbance is linked to the formation of aromatic dimers and functionalization reactions. Hems et al. report that the absorption enhancement was observable up to 11 equivalent hours of $OH \cdot$ exposure, and diminished with a net exposure of up to 42 hours. It should be noted that the samples analyzed by Hems et al. was only the water-soluble portion, obtained by gathering BrC on a filter and





extracting with purified water. Furthermore, the OH· oxidation was carried out by adding hydrogen peroxide to the

extract solution and exposing it to UV lamps. The OH· oxidation performed by the PAM during FIREX is far less

controlled, given the dynamic nature of the plumes we sampled.

Figure 7 shows $E_{MAC}(\lambda)$ at 488 and 561 nm for all OH· experiments. The data is summarized in Table 3.

**[Figure 7]**

While $OH_{EXT}$ values were large, it is noteworthy that they were typically exceeded by $NO_{3,EXT}$ values, which, as noted

in the previous section, indicates the presence of BBVOCs that are more reactive toward $NO_3$· than toward OH·. The

exact nature of these sensitivities requires further research.

**[Table 3]**

The same scatterplot analysis was applied to the OFR_1 step (refer to Fig. S4) of the $OH_{Arizona}$ experiment, however,

it is less illustrative than with $NO_3$· because the primary driver of $MAC(\lambda)$ diminishment is fragmentation reactions.

Before this step, $OH_{EXT}$ was 74.95 $s^{-1}$ and after, it was 59.37 $s^{-1}$, giving an approximate equivalent age of 18.27 ± 7.54

days. The enhancement ratios of the individual ion families show a decrease in $CHO_1N$ and an increase in $CHO_{gt1}N$,

though to a lesser degree than under $NO_3$· aging. The ambient $CHO_{gt1}N$ mass fraction was approximately 25% higher

in the $NO_{3,Arizona}$, as well – the comparatively lower relative abundances of nitroaromatics during $OH_{Arizona}$ (both pre-

and post-OFR_1) may obfuscate any meaning in the enhancement ratios.

Figure S7 shows the $OH_{Arizona}$ scatterplot with markers, and Fig. S8 shows the individual *m/z* measured by the SP-

AMS. Qualitatively, it can be observed that there is less spread in the scatterplot and the points are grouped closer to

the 1:1 line, further suggesting that the dominant mechanism is fragmentation.

Figure 8 shows the ion family enhancement ratios through the same OFR_1 step corroborating the conclusions drawn

from the scatterplots.

**[Figure 8]**

## 4 Conclusions and Future Work - Synthesizing Daytime and Nighttime Aging

The observations of $E_{MAC}(\lambda)$ and the associated chemistry from FIREX-AQ represent the first attempt to use an OFR

in a mobile setting to sample biomass burning at their source, as well as the first application of the novel MIPN v1 to

a field study. Observations track closely with previous laboratory studies.

These results show the difficulty in naïvely applying a particular aging model to atmospheric aerosol to constrain their

long-term behavior in climate models. Aerosol does not age along any single pathway for more than half of a diurnal

cycle: at night, oxidative aging of BrC by $NO_3$· increases MAC, whereas daytime oxidative aging by OH· decreases

MAC. Overall, our results suggest that explicit characterization of the effect of diel aging on atmospheric aerosol

optical, chemical, and physical properties represents the best possible input to climate models.





**Data availability.** All experimental data, including MIPN; SP-AMS; SP2; and Vocus PTR-MS measurements, rate constants, and data tables used to calculate external reactivities are available for download at
doi:10.17632/5mr43vbks3.1 (Sumlin, 2021).

**Supplement.** Includes eight figures detailing rBC fraction observed during experiments (S1 and S2), sample timelines for oxidation experiments (S3 and S4), and AMS signal enhancements for various ion families upon oxidation (S5 through S8). Also includes two tables (S1 and S2) with fit coefficients for Eqs. (1) and (2).


**Author contributions.** RKC and BS conceived of the study and its design. RKC and AL provided guidance and supervision for carrying out the research tasks, interpretation of results, and contributed to the preparation of the manuscript. BS, EF, NJS, FM, CD, and SH performed the experiments. BS performed the data analysis, developed the figures, and led the preparation of the manuscript. All authors were involved in the editing and proofreading of the
manuscript.

**Competing interests**. The authors declare that they have no conflict of interest

**Acknowledgements**

The authors would like to acknowledge Tara Yacovitch, Rob Roscioli, Jordan Krechmer, and Tim Onasch of Aerodyne
Research, Inc. and Art Sedlacek III of Brookhaven National Laboratory for their effort in processing and contributing AML data to the public FIREX-AQ repository. Chakrabarty and his group acknowledges support from the US National Science Foundation (AGS-1455215 and AGS-1926817) and the US Department of Energy Atmospheric System Research program (DE-SC0021011). Sumlin acknowledges support from NASA Earth System Science Fellowship (80NSSC18K1414).






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



**Figures and Tables**

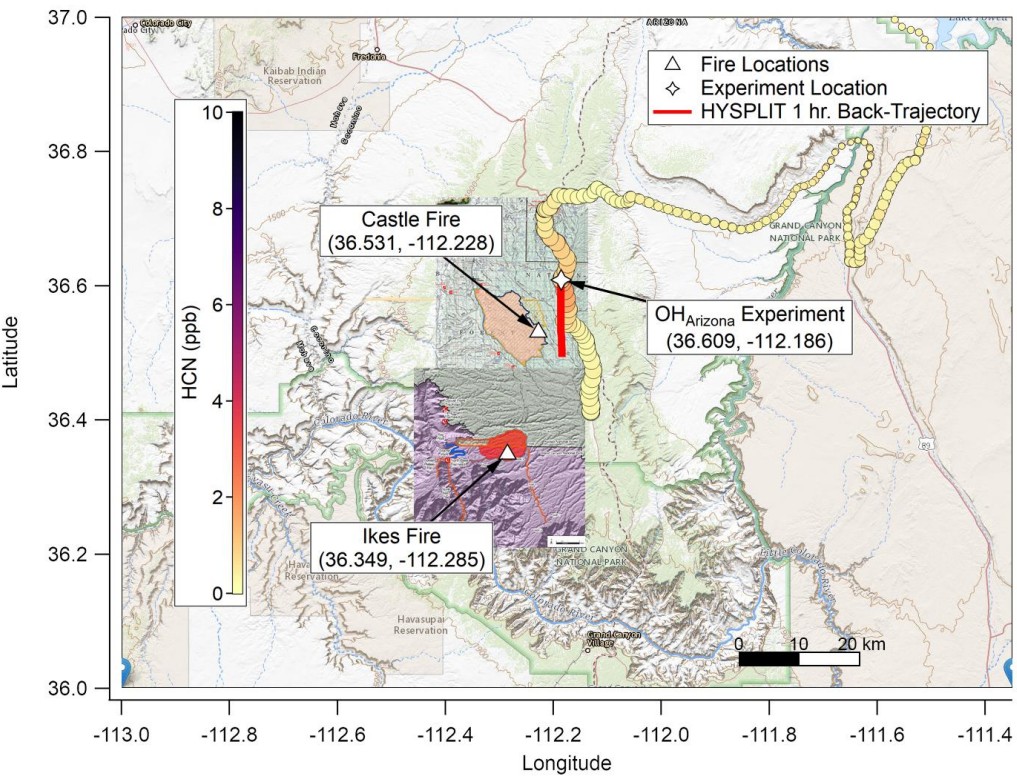

Figure 1: Map of the Grand Canyon region overlaid with HCN concentration (ppb, color scale), elevation (HCN marker size, larger is higher AMSL), fire origin locations from Incident Information System (white triangles), OH$_{Arizona}$ experiment location (white star), and fire boundary maps for 08/20/2019. Based on HYSPLIT back-trajectories, the plume for this experiment was less than one hour old, though surface concentrations of smoke were comparatively lower since the nocturnal boundary layer had lifted. Background map is from USGS EarthExplorer, and fire map overlays are from the National Wildfire Coordinating Group's Incident Information System (InciWeb).



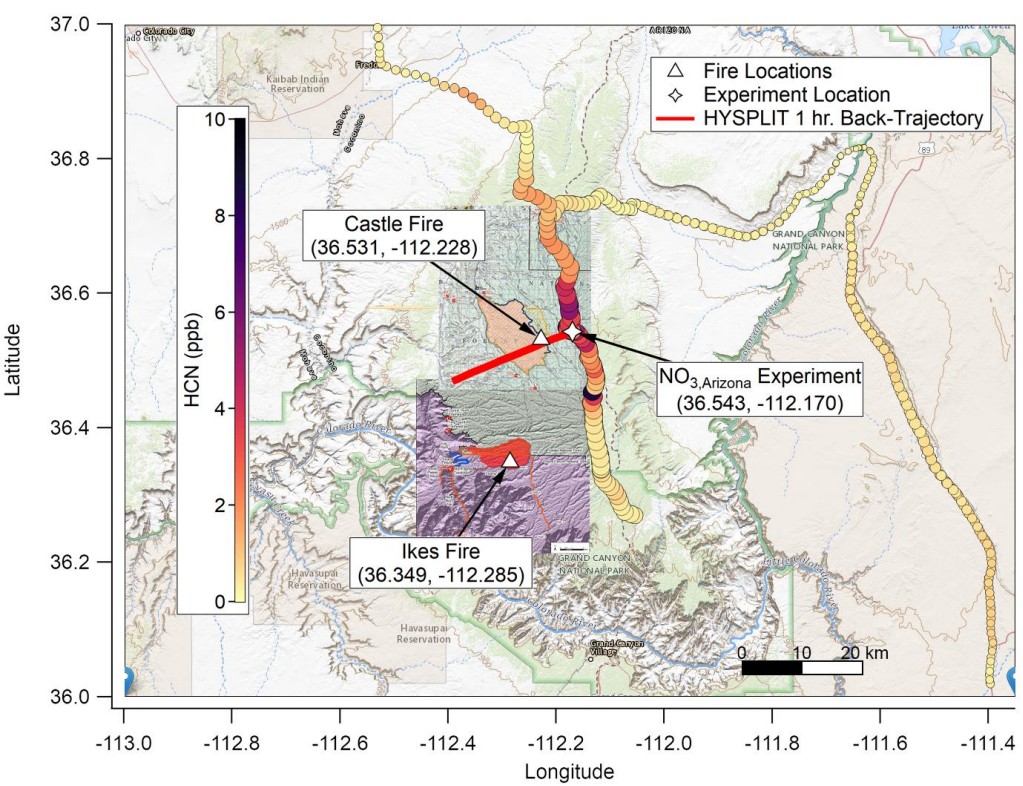

**Figure 2: As with Figure 2, but for NO₃,Arizona on 08/22/2019. Based on HYSPLIT back-trajectories, the plume for this experiment was again less than one hour old, though smoke that had settled in the valley under the nocturnal boundary layer may have been older. Background map is from USGS EarthExplorer, and fire map overlays are from the National Wildfire Coordinating Group's Incident Information System (InciWeb).**




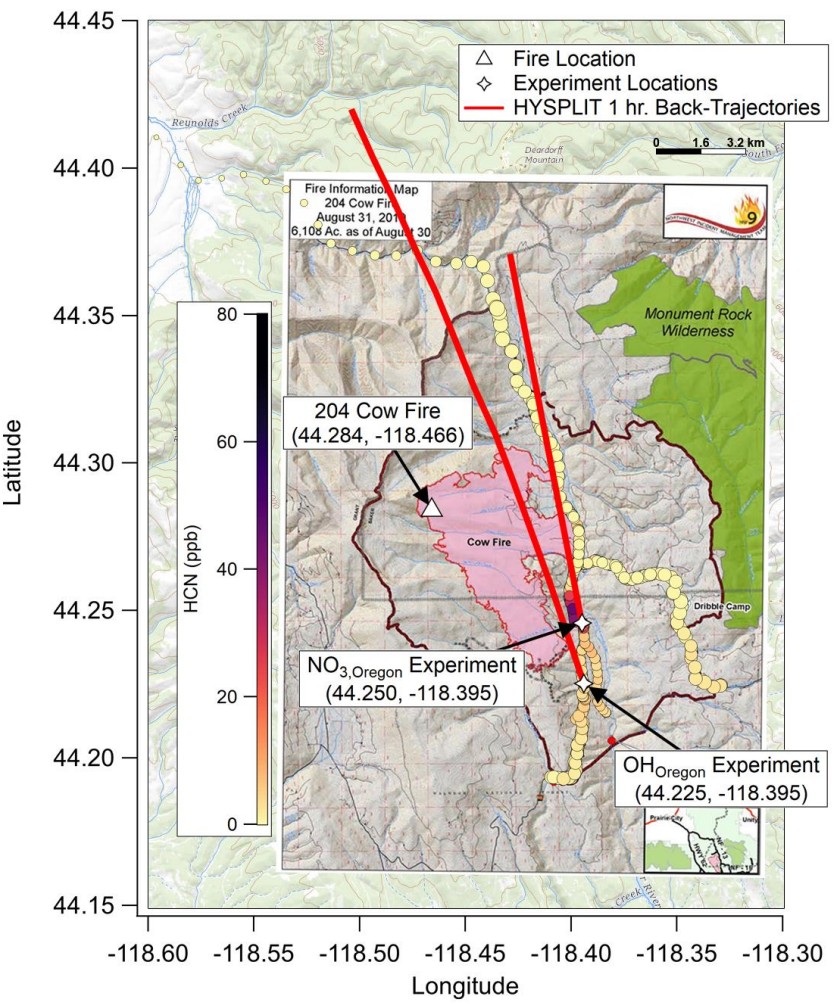

**Figure 3: As with Figures 2 and 3, showing fire ignition point (white triangle), OH_Oregon and NO3_Oregon experiment locations (white stars), and fire boundary maps for 08/26/2019. HYSPLIT back-trajectories again show the plumes for these experiments were much less than one hour old. Background map is from USGS EarthExplorer, and fire map overlays are from the National Wildfire Coordinating Group's Incident Information System (InciWeb).**






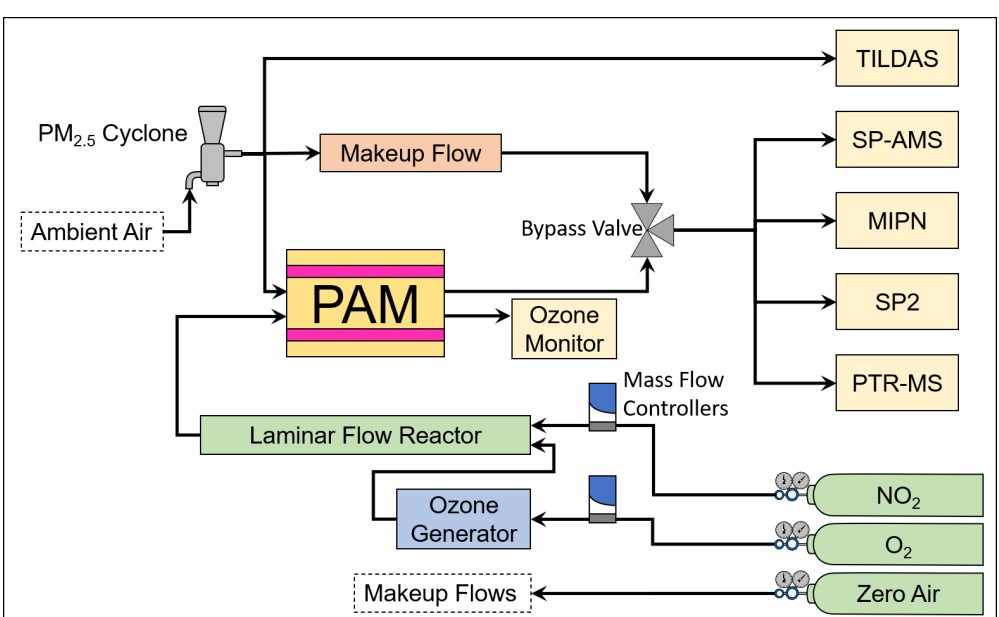

**Figure 4: A schematic of the general experimental setup aboard the AML showing instruments, reactors, reagents, and other hardware.**


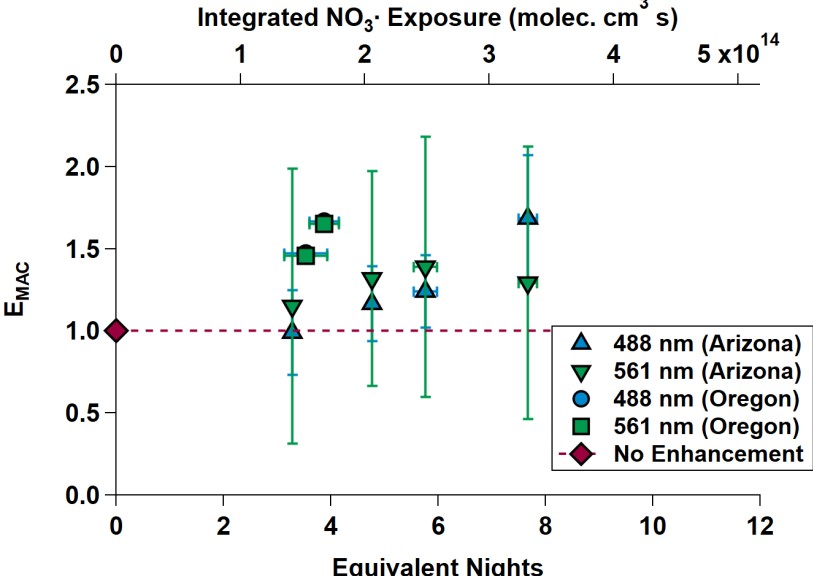

**Figure 5: MAC(λ) Enhancements at 488 and 561 nm during NO$_{3,\text{Arizona}}$ (triangles) and NO$_{3,\text{Oregon}}$ (square and circle; the circle is partially obscured by the square). The gray dashed line is the "no enhancement" line. The Arizona experiment data shows an increasing trend in MAC(λ) with increasing equivalent age. The Oregon data on its own does, as well, but in**





context of the Arizona data, the chemical species present in the smoke plume in Oregon appear to have been much more reactive towards $NO_3\cdot$. Equivalent photochemical age is calculated from the $NO_3\cdot$ exposure in the OFR and assuming an average ambient $NO_3\cdot$ concentration of $5\times10^8$ molec $cm^{-3}$ (Atkinson et al, 1991).

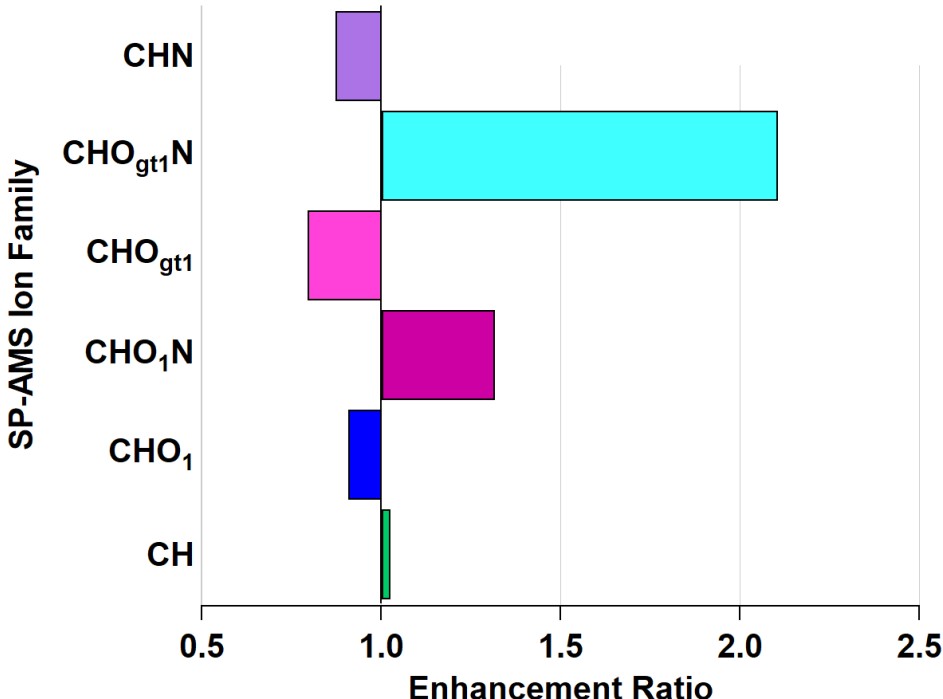

**Figure 6:** Enhancement and depletion of ion families measured by the AMS during $NO_3\cdot$ oxidation. $CHO_1N$ and $CHO_{gt1}N$,
while present in lower relative abundances, showed the largest enhancement through $NO_3\cdot$ oxidation.

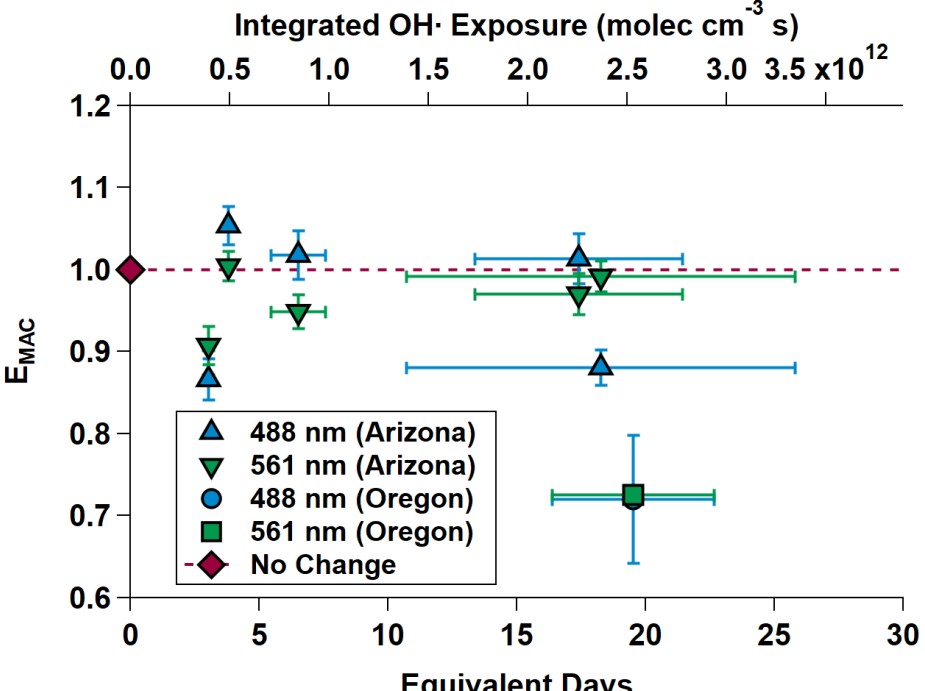

**Figure 7:** MAC(λ) Enhancements at 488 and 561 nm during OH$_{Arizona}$ (triangles) and OH$_{Oregon}$ (square and circle, which is partially obscured). The gray dashed line is the "no enhancement" line. The shows an initial diminishment, then subsequent increasing to decreasing trend in MAC(λ) with increasing equivalent age. Equivalent photochemical age (days) is calculated from the OH· exposure in the OFR and assuming an average ambient OH· concentration of $1.5 \times 10^6$ molec cm$^{-3}$ (Mao et al, 2009).


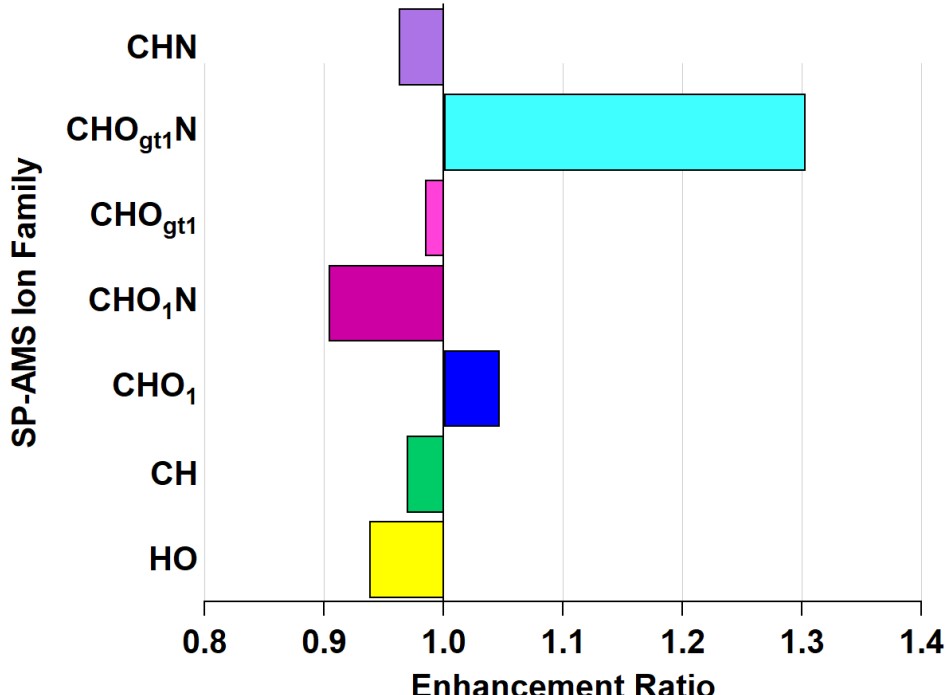

**Figure 8: Enhancement and depletion of ion families measured by the AMS during OH· oxidation. $CHO_1N$ is depleted while $CHO_{gt1}N$ is enhanced, however, both these families are present in low relative abundances which may exaggerate the enhancement ratio.**




Table 1: Study-average values of mass absorption coefficients (MAC, m² g⁻¹) from Arizona and Oregon. Errors are one standard deviation.

| Wavelength | Arizona MAC | Oregon MAC |
|---|---|---|
| 488 nm | $1.93 \pm 0.33$ | $1.95 \pm 0.41$ |
| 588 nm | $1.71 \pm 0.32$ | $1.72 \pm 0.05$ |


Table 2: $E_{MAC}$(488 nm) and $E_{MAC}$(561 nm) for $NO_3\cdot$ oxidation experiments. Asterisks denote $NO_{3,Oregon}$. All others taken during $NO_{3,Arizona}$, and the dagger indicates $O_3$-only oxidation.

| Equivalent nights | $E_{MAC}$ at $\lambda = 488$ nm | $E_{MAC}$ at $\lambda = 561$ nm |
|---|---|---|
| $3.28 \pm 0.00$ | $0.99 \pm 0.26$ | $1.15 \pm 0.79$ |
| $3.53 \pm 0.28$* | $1.47 \pm 0.01$ | $1.46 \pm 0.01$ |
| $3.87 \pm 0.40$* | $1.67 \pm 0.01$ | $1.65 \pm 0.01$ |
| $4.76 \pm 0.00$ | $1.17 \pm 0.23$ | $1.32 \pm 0.66$ |
| $5.76 \pm 0.22$ | $1.24 \pm 0.22$ | $1.39 \pm 0.79$ |
| $7.67 \pm 0.17$ | $1.69 \pm 0.38$ | $1.29 \pm 0.84$ |
| n/a[†] | $1.06 \pm 0.96$ | $1.05 \pm 0.94$ |


Table 3: $E_{MAC}$(488 nm) and $E_{MAC}$(561 nm) for $OH\cdot$ oxidation experiments. Asterisk denotes $OH_{Oregon}$. All others taken during $OH_{Arizona}$.

| Equivalent days | $E_{MAC}$ at $\lambda = 488$ nm | $E_{MAC}$ at $\lambda = 561$ nm |
|---|---|---|
| $3.03 \pm 0.20$ | $0.88 \pm 0.02$ | $0.94 \pm 0.02$ |
| $3.79 \pm 0.05$ | $1.05 \pm 0.02$ | $1.00 \pm 0.02$ |
| $6.51 \pm 1.06$ | $1.02 \pm 0.03$ | $0.95 \pm 0.02$ |
| $17.41 \pm 4.04$ | $1.02 \pm 0.03$ | $0.97 \pm 0.03$ |
| $18.27 \pm 7.54$ | $0.88 \pm 0.02$ | $0.99 \pm 0.02$ |
| $19.52 \pm 3.14$* | $0.72 \pm 0.08$ | $0.73 \pm 0.01$ |
