# Peer review of "Diel Cycle Impacts on the Chemical and Light Absorption Properties of Organic Carbon Aerosol from Wildfires in the Western United States"

_Atmospheric Chemistry and Physics, 2021_

## Author Comment (AC1)

We are thankful for the anonymous reviewers' feedback on our manuscript and are happy to further the discussion by sharing our responses. Where appropriate, we have incorporated the suggestions into our manuscript, and look forward to publication in Atmospheric Chemistry and Physics.

In this document, the reviewer comments are in plain text, our responses are in blue, and when necessary, we have included the updated language in the manuscript in orange.

**Response to RC1**

This paper describes absorption measurements for wildfire smoke in Oregon and Arizona using photoacoustic spectroscopy. The smoke is aged in a oxidation flow reactor using OH or NO3 for the equivalent of ~5-20 days. The number of measurements are very limited (2 examples of aging with OH and 2 examples of aging with NO3), but the approach hasn't been reported for field measurements. The paper is short and well-written. I recommend publication after the following comments are addressed.

- Line 55: Warneke 2018 is not published. Change to "in preparation".

This change has been made on line 59 and in the reference list.

- Section 2.1.1 and 2.1.2: Add the source for the vegetation identifications.

Vegetation information was acquired from a variety of sources, including visual identification of species while we were physically in the forests. A complete description of the fuel bed would be too extensive for this manuscript, and we refer the reviewer and readers to Appendix A of Riccardi et. al. 2007, The fuelbed: a key element of the Fuel Characteristic Classification System, Canadian J. of Forest Res., 37,12, which contains an exhaustive list of all fire-relevant flora species categorized by region in North America. This citation is added to the revised manuscript.

- Section 2.1.1 and 2.1.2: What were the average wind speeds during each of the four measurements?

Surface wind speeds for the Arizona measurements averaged 2.98 m s-1 during the OH experiment, and 0.39 m s-1 during the NO3 experiment. In Oregon, surface winds were an average of 0.43 m s-1 during the OH experiment, and 1.40 m s-1 during the NO3 experiment. These values have been added to the manuscript where appropriate.

- Section 2.1 is named "Wildfire emissions sampled during OFR experiments". It would make sense to move the final paragraph (lines 94-114) to Section 2.2 ("Instrumentation"). That paragraph introduces the Aerodyne Mobile Laboratory and fits more logically with instrumentation than with the description of the fires.

Agreed. This change has been made to the manuscript.

- Line 156: How accurately did the UV flux measurement at the exit flange represent the UV flux inside the OFR? How did you evaluate that?

UV flux is measured by the photodiode positioned inside the OFR near the exit flange, not within the exit flange. The photodiode has an unobstructed view into the OFR.

- Line 160: The calculated mean residence time of the oxidation flow reactor was 123 s. What was the observed residence time? If the OFR was abruptly filtered or overflowed with zero air, what was the observed residence time to fully flush the volume? Add this value as well.

Strictly speaking, no *measurements* of these times were taken during the field experiment. The reactor was never simply filtered nor abruptly filled with zero air – all cleanout procedures were also accompanied with activating the UV lamps to full power. This confounds the traditional metrics of residence time and cleanout time. We typically think of cleanout time as the amount of time necessary to reduce a tracer concentration by three e-folding times, or the time it takes to reduce tracer concentration to approximately 5% of starting concentration. During these field studies, we adopted 300 ng m-3 of organic aerosol as a "clean" baseline concentration between experiments. The time it took to achieve this was never recorded due to the complexities of managing the AML with limited crew, however, no experiments were begun unless this threshold was met.

The observations of these parameters are better left to controlled laboratory studies. The reviewer's question regarding residence times in a PAM OFR are addressed in the literature. See, for example, Mitroo et al. (2018). This reference has been added to the manuscript at line 164. For relevant changes to the text, please see our response to the next comment.

- Figures S3 and S4 in the SI material show that the OH and NO3 reagent concentrations were always changed monotonically. They began at the highest concentration, and then were decreased sequentially in steps. If the ambient aerosol properties are changing as a function of time, this approach risks confusing that change as a dependence on the OH or NO3 concentration. It may also introduce an error if there are systematic measurement drifts. It would be better in the future to both 1) Introduce different reagent concentrations in random order; 2) Repeat one or two reagent concentrations to examine the repeatability of the measurements. It would be worth noting these in the paper as future approaches.

This is an excellent point to consider when conducting OFR experiments, and the reviewer proposes two satisfactory approaches to addressing this issue. Our approach to mitigating these errors was to bypass the OFR and measure ambient air before every change in OFR conditions, and to only compare any given reaction step to the ambient measurements immediately before and after. This is, however, not the only approach and we appreciate the suggestion for future experiments. We further acknowledge that the language in section 2.3 does not adequately describe the experimental procedure. The relevant text in section 2.3 now reads (including our response to the previous comment):

The total instrument plus makeup flowrate through the OFR was 6.4 liter min-1, corresponding to a calculated mean plug flow residence time of 123 sec. The degree to which plug flow is a valid model for an OFR is the subject of ongoing research, however, this assumption is adequate for the flowrates and experimental setups used here (Mitroo et al. 2018). OFR experiments lasted 1 to 2 hours, during which the reagent inputs were controlled in steps to

simulate varying degrees of atmospheric aging. An experiment step consists of two phases, an ambient phase and a reaction phase. During the ambient phase, instruments first sampled ambient air for 5-10 min (depending on the experiment timeline) while LFR/OFR conditions equilibrated for the reaction phase. During the ambient phase, 5 L min-1 of makeup flow was pulled through the OFR to reduce stagnation time. Once the ambient phase ended, an electronically actuated 3-way valve was switched to connect the instruments to the OFR, and OFR-processed air was sampled for 5 to 10 min. After each step, the OH· or NO3· exposure was changed, and the above measurements were repeated.

- Line 170-172: The word "steps" is awkward. Consider changing "steps" to "measurements." Also, it is confusing that oxidation steps are labeled as "OFR\_OH\_X" and "NO3PAM\_O3". The OFR and the PAM are the same thing. It would be better to name these consistently (not with some labels as PAM and some as OFR).

For consistency with the previous comment, we have chosen to use the term "phase" rather than "steps". Furthermore, we have standardized all labels to "OFR" in figures and the text.

- Throughout the paper, NO3 and OH equivalent exposures are given in units of "days" or "equivalent days". Since OH is present during the day and NO3 is present at night (so that each equivalent day would take ~2 days in the ambient atmosphere), it may be clearer to give the exposures in "hours" or "equivalent hours".

The community seems to have widely adopted "days" as the unit of choice, at least when discussing OH oxidation, and this convention has propagated to "nights" when discussing NO3 oxidation. However, we agree with this comment, especially when considering our conclusions that aging along one oxidative pathway for more than half a diurnal cycle (12 hours) may be unrealistic. Where applicable, we have rewritten our units in terms of hours, and included days or nights as a parenthetical in the text at the beginning of sections 3.2 and 3.3. Tables 2 and 3 now report only hours.

- Line 230: It would be useful to specify that the AAE values calculated here are for both BC and BrC.

We are confident that the calculated AAE values corresponded to light absorbing OA constituents in the intercepted plumes. This is because of the negligible amount of rBC measured is likely an artifact from charred organics erroneously reported by the SP2 (Sedlacek et al., 2018, Formation of refractory black carbon by SP2-induced charring of organic aerosol, Aerosol Sci. & Tech, 52:12, 1345-1350, DOI: 10.1080/02786826.2018.1531107.

On line 240, we have added:

As shown in Figures S1 and S2, the rBC mass fraction in the sampled plumes were negligible (<5%) and within the margin of error due to charring of OA in the SP2 (Sedlacek et al, 2018). Therefore, we are confident that the calculated AAE values corresponded to light absorbing OA constituents in the intercepted plumes.

- Section 3.2: This section seems to introduce several results without explanation. Specifically:

- Line 239: NO3,EXT is not defined. What is it and how was it measured or calculated?

 $NO_{3,EXT}$  (and similarly,  $OH_{EXT}$ ) is the total summed reactivity of ambient gas-phase BBVOCs to  $NO_3$  (OH). Concentrations are measured by the PTR-MS during the ambient phases and calculated by  $NO_{3,EXT}$  or  $OH_{EXT} = k_{NO3 \text{ or } OH} \times [NO_3 \text{ or } OH] \times [BBVOC]$ . We have clarified this in the manuscript at line 252. All supplemental data files (including an Excel spreadsheet of 260 BBVOCs, their measured concentrations, relevant rate constants, and reactivities for all the OFR experiment ambient measurements) are available at doi:10.17632/5mr43vbks3.1, as specified in the "Data Availability" section.

- Line 240: The NO3 reactivity of catechol is introduced, but there was no description of an instrument to measure gas-phase VOCs. How was this measured and calculated? There is no catechol data shown in figures or tables.

Catechol (and other gas-phase VOCs) were measured by a Vocus Proton Transfer Reaction Mass Spectrometer, which was mentioned at the end of section 2.1.2, however, it has been moved to section 2.2 per an earlier comment. We have clarified that the measurement was obtained by PTR-MS in section 3.2.

Since the reactivity of each individual VOCs is not germane to the conclusions of this paper, the observation of catechol by the PTR-MS is included only to emphasize the high ambient concentrations of BBVOCs that are particularly reactive to NO3. See Finewax et al. 2017, for example. This reference has been added to the manuscript on line 257.

- Line 243: "The difference in external reactivities manifested in different EMAC( $\lambda$ ) behavior at the two sites." What does external reactivities mean and where are they shown?

We have added language clarifying what is meant by "external reactivity" on lines 252-253. We measured some 260 individual BBVOCs during these experiments and this would be unwieldly to include in a table in the manuscript. Per the previous comments, the Excel spreadsheet of BBVOC reactivities is available at doi:10.17632/5mr43vbks3.1, as specified in the "Data Availability" section.

- The notation CHO1N and CHOgt1N are unclear. I suggest making two changes: 1) use x to indicate that molecules may contain multiple atoms of C, H, and N; 2) change gt1 to >1. For example: CxHxO1Nx and CxHxO>1Nx.

The software package used to process the AMS data characterizes ions according to their families, and assigns masses to a family from among  $C_X$ , CH, CHO1, CHOgt1, CHN, CHO1N, CHOgt1N, CS, HO, NH, Cl, NO, CO, Air, Tungsten, and Other. This nomenclature is ubiquitous in the literature and immediately recognizable to the AMS and OFR community. We have elected to leave this terminology in place. However, to clarify what is meant by those family names, on lines 285-289 we have made the following modification:

The CHOgt1N ("Ogt1" indicates more than one oxygen atom in a molecule containing one or more atoms of C, H, and N) family is enhanced by 211% and CHO1N ("O1" indicates a single oxygen atom in a molecule containing one or more atoms of C, H, and N) by 132% through

the OFR, with a corresponding diminishment in  $CHO_{gt1}$  and  $CHO_1$  (where, again, " $O_{gt1}$ " and " $O_1$ " indicate the quantity of oxygen atoms in molecules of one or more atoms of C and H) to 79% and 91%, respectively.

**Response to RC2**

**General Comments:**

This manuscript investigates the dynamics of biomass burning aerosol optical properties (MAC) under different oxidation schemes. This authors pursue this investigation through use of a well-equipped mobile lab and oxidation flow reactor, which allows near in-situ sampling, aging, physical characterization, and chemical characterization of the target aerosol. Through investigating the plumes of two fires and their aging under two oxidation schemes the authors draw conclusions about how aging will change the optical properties of the observed aerosol. The authors conclude that nighttime (NO3.) aging generally leads to more absorbing aerosol and daytime (OH.) aging has more complex effects, which trend toward less absorbing aerosol for longer aging times.

Overall this short paper is well written, investigates a topic that is of pressing interest to the atmospheric community, and presents a unique application of oxidation flow reactors. The accurate constraints on aerosol optical properties, and how they change as they age, are key to accurate estimations of the earth's radiative budget and modeling of future climate systems. I recommend that this work be accepted once the minor issues I've elaborated on below have been addressed.

**Specific Comments:**

Line 12: These statements seem to over simplify the results. I recommend slightly restructuring the abstract to better present the detailed results the authors discuss in the manuscript. For example, reading the abstract only, I would believe that OH oxidation was only observed to be bleaching, whereas the authors observed a much more complex relationship. Similarly, the NO3 results were aging and fire dependent. This context should be included here if possible.

We did indeed observe a more complicated relationship than the abstract suggests. We have rewritten lines 12-18 of the abstract as follows:

We found that  $OH^{\cdot}$  exposure induced a slight initial increase in absorption corresponding to short timescales, however, in longer time scales, the wavelength-dependent MAC( $\lambda$ ) decreased by a factor of 0.72  $\pm$  0.08, consistent with previous laboratory studies and reports of photobleaching. On the other hand, NO3 exposure increased MAC( $\lambda$ ) by a factor of up to 1.69  $\pm$  0.38. We also noted some sensitivity of aerosol aging to different fire conditions between Arizona and Oregon. The MAC( $\lambda$ ) enhancement following NO3 exposure was found to correlate with an enhancement in CHO1N and CHOgt1N ion families measured by an Aerodyne aerosol mass spectrometer.

Line 246: Can the authors comment on the differences between the BBVOCs that drove these differences in MAC changes? The VOCUS data is used to investigate potential differences in the NO3 reaction rate, but it would be also interesting to use this data to investigate particular precursors which contribute strongly to the observed enhancements/bleaching.

The reactivities of individual BBVOCs and the differences therein are beyond the scope of this manuscript, whose primary purpose is to communicate the changes in optical properties of forest fire emissions. However, to facilitate such an investigation, an Excel spreadsheet of all 260 BBVOCs measured during the experiments, and, where applicable, their rate constants with OH or NO3 and their calculated reactivities is available at doi:10.17632/5mr43vbks3.1, as specified in the "Data Availability" section.

**Technical Comments:**

Line 58: Line suggests investigations regarding the refractive index of the sampled aerosol, but the paper only includes mentions of MAC data.

We have changed the sentence on line 60. It now reads:

Results are used to quantify the effects of diurnal cycle-driven oxidation processes on the OA mass absorption cross-section.

Line 112: Small font error in reference

Corrected.

Line 239: Need to better define NO3, EXT class of variables. I think they are, are related to, the NO3,REXT, but it is not immediately clear.

Please see our response to several prior comments. We have further clarified what is meant by "external reactivity" and included supplemental information. Furthermore, "NO3,REXT" is a typo. We thank the reviewer for bringing that to our attention.

---

## Author Response (AR2)

To the editor:

Thank you for the opportunity to publish in *Atmospheric Chemistry and Physic*s. We are also grateful for the comments and questions from the two anonymous reviewers. Their input has strengthened our manuscript and we are happy for the opportunity to share our work. This final submission contains corrections made at your request. Please find our response below in blue text, and we have also included the relevant language change from the manuscript in orange.

We note that there may be a discrepancy between the line numbers provided in these comments and the line numbers in our working document. For example, the comment about "writ large" on line 301 refers to line 281 in our document. We have made our best effort to identify the location of the comment.

- Line 120: The Abstract, this line and table 1 each indicate a different wavelength (561 nm, 532 nm, and 588 nm). Please check that the correct wavelength is used throughout the paper.

    This correction has been made and the manuscript now refers only to the correct wavelength of 561 nm.

- Line 132: delete 'enhancement' at the end of this sentence

    This correction has been made. Lines 116-117 now read:

    Thus, $E_{MAC}(\lambda) > 1$ indicates an oxidative aging-induced absorption enhancement, while $E_{MAC}(\lambda) < 1$ indicates diminished absorption.

- Line 289-290: include the units for the ambient MAC values of this study

    The values given in the text in section 3.1 are for ambient AAE, which is unitless. We have made the following change to line 235:

    Table 1 summarizes the average $MAC(\lambda)$ values ($m^2\ g^{-1}$) of ambient aerosol obtained from Arizona and Oregon.

- Line 301: I don't think "writ large" fits here

    This correction has been made. Lines 280-281 now read:

    While knowledge of the individual *m/z* enhancement and depletion may inform future investigations, it is perhaps more illustrative to consider ion families.

- Conclusions: The same way that the abstract was modified to reflect the dynamic influence of OH on MAC, the conclusions should be change to correctly indicate that OH first increased MAC, but at longer oxidation times, bleaching was observed

    This correction has been made. Lines 328-330 now read:

    Aerosol does not age along any single pathway for more than half of a diurnal cycle: at night, oxidative aging of BrC by $NO_3\cdot$ increases $MAC(\lambda)$, whereas daytime oxidative aging by $OH\cdot$

initially increases MAC($\lambda$), which is followed by a strong reduction in MAC($\lambda$) due to bleaching.

---

## Author Response (AR3)

We have made the correction on line 269 of the manuscript and units are now included.